# Effect of Vacuum Roasting on Total Selenium Content of Selenium-Enriched Rapeseed, Maillard Reaction Products, Oxidative Stability and Physicochemical Properties of Selenium-Enriched Rapeseed Oil

**DOI:** 10.3390/foods12173204

**Published:** 2023-08-25

**Authors:** Qihui Xie, Chengming Wang, Luqiu Peng, Yiyang Dong, Yu Gao, Jing Xu, Hongzheng Ping, Shilin Liu

**Affiliations:** 1College of Food Science and Technology, Huazhong Agricultural University, Wuhan 430070, China; 2Key Laboratory of Environment Correlative Dietology, Huazhong Agricultural University, Ministry of Education, Wuhan 430070, China

**Keywords:** selenium-enriched rapeseed, vacuum, roasting, selenium loss, Maillard reaction, lipid oxidation, oxidative stability

## Abstract

Selenium-enriched rapeseed (SER) is an emerging oil seed. Roasting is beneficial in improving oil yield and promoting the release of micronutrients into SER oil, but high temperatures and dry air lead to selenium loss and fatty acid degradation in SER. To minimize the selenium loss and improve the SER oil quality, this study investigated the effects of vacuum (VC) roasting (90–170 °C for 30 min) on the SER selenium content, Maillard reaction products, oxidative stability, and physicochemical properties of SER oil, with conventional dry air (DA) roasting as the control. The results showed that the selenium loss in VC-roasted SER meals increased from 7.17 to 19.76% (90–170 °C for 30 min), which was 47.13 to 80.48% of that in DA-roasted SER meals, while no selenium was detected in the SER oils. Compared to DA roasting, VC roasting (90–170 °C for 30 min) reduced lipid oxidation products (LOPs), Maillard reaction products (MRPs), and benzo[a]pyrene contents, and increased carotenoids, unsaturated fatty acid contents, reaching a maximum oil yield of 35.58% at a lower temperature (130 °C for 30 min). Selenium contents exhibited a highly significant negative correlation with MRPs and LOPs (*p* ≤ 0.005). The VC roasting retarded selenium loss and improved SER oil quality compared to conventional DA roasting.

## 1. Introduction

Rapeseed is the world’s second major oilseed crop after soybean. Rapeseed oil is rich in monounsaturated and polyunsaturated fatty acids, phenols, tocopherols, and carotenoids [1]. Besides being used as a source of oil, rapeseed is a high-quality source of protein, with crude protein content reaching 30–40% [2]. Selenium-enriched rapeseed (SER), with selenium content exceeding 20 μg/kg according to the local standard DB61T556-2018 [3] of Shaanxi Province, China, is an emerging agricultural product that is of great interest to researchers due to its high selenium content. Selenium (Se) is an essential trace element and a crucial component of selenium-containing proteins, including glutathione peroxidases (GPxs), thioredoxin reductase (TrxR), and t-RNA. The adequate intake of selenium can improve immunity, prevent cardiovascular disease, and combat oxidative stress [4]. SER contains up to tens of times more selenium than regular rapeseed, making it highly valuable for nutritional and medical purposes.

Roasting reduces the moisture content of rapeseed and alters the structure of the proteins, thereby enhancing the oil yield of the rapeseed, promoting the release of trace nutrients, such as carotenoids, tocopherols, phenols, etc., imparting a distinctive flavor to the oil, and improving the oxidative stability of the rapeseed oil [5]. Dry air (DA) roasting is a conventional pre-press heat treatment method for rapeseed, where rapeseeds are roasted in an oven using dry hot air. This treatment is used frequently for its ease of implementation and low operating costs [6]. However, Se loss may occur during DA roasting because the seleno-amino acids in SER can react with the reducing sugars (Maillard reaction) at high temperatures to form volatile Se compounds [7], and their oxidation potential is lower than that of thioamino acids, making the seleno-amino acids more susceptible to oxidative degradation as volatile Se products [8], which limits the further utilization of SER as by-products in the selenium-enriched food industry. On the other hand, the high content of unsaturated fatty acids in SER is more susceptible to auto-oxidation under high temperature conditions, leading to a decrease in the quality of fatty acids and an increase in undesirable products, such as free fatty acids, peroxides, carbonyl compounds, and conjugated olefins, in the oils [9]. Therefore, there is a need to develop an effective method to reduce the loss of Se in SER and mitigate the oxidation of SER oil.

Different to DA roasting, vacuum (VC) roasting reduces dry air convective heat transfer and isolates oxygen, which is more effective in mitigating thermal oxidation and removing moisture from materials due to the low boiling point of water. Meanwhile, previous studies by Gulcan et al. had found that inert atmosphere roasting significantly reduced the formation of Maillard reaction products (MRPs), such as acrylamide and 5-hydroxymethylfurfural, in bread [10], while Anese et al. also reported that VC roasting reduced the formation of acrylamide in coffee beans compared to DA roasting [11], indicating that the Maillard reaction is inhibited in an anoxic atmosphere. Moreover, an earlier study by Rattana et al. also showed that VC roasting reduced the peroxide value and increased the unsaturated fatty acid content of black sesame seed oil compared to DA roasting [12]. As a result, we attempted to minimize both the oxidation and the Maillard reaction using VC roasting, aiming to preserve more of the Se and reduce the oxidation products of the SER.

Apart from Se loss, previous research on the effects of vacuum roasting on oil seed has focused on drying performance and energy consumption [13], and there is a lack of literature available regarding the effect of VC roasting on the quality of SER oil compared to conventional DA roasting. In this study, we aimed to use VC roasting to retain more Se and improve the quality of the SER oil, as well as confirming the correlation between selenium loss and the Maillard reaction, lipid oxidation.

## 2. Materials and Methods

### 2.1. Materials

SER originated from a natural selenium-enriched zone in Ankang, Shanxi Province, China. The seeds were harvested in September 2021, purchased from Ankang Selenium Source Oil Group Co., Ltd., and stored at 4 °C in a refrigerator. Selenium, 5-hydroxymethylfurfural, and heptadecanoic acid methyl ester standards were purchased from Yuanye (Shanghai, China), while the benzo[a]pyrene standard was purchased from Anpel (Shanghai, China). All chemicals used in the determination of the Se content were guaranteed reagents, and those used in gas chromatography and liquid chromatography were chromatography-grade reagents.

### 2.2. Roasting and Oil Extraction

The iron tray (20 × 25 cm) was preheated in a vacuum drying oven (DZF-6020A, LICHEN, Changsha, China). Once the specified temperature was reached and stabilized, the SER (110 g) was placed in the tray and roasted for 30 min under vacuum (−0.98 MPa) and dry air conditions at temperatures of 90, 110, 130, 150, and 170 °C, respectively. After roasting, the SER was quickly cooled to −20 °C for 5 min. The seeds were then pressed at the lowest set temperature (62 °C) using a small laboratory oil press (BOZY-01G, HANHUANG, Yueqing, China). The oil was centrifuged (8000 rpm, 15 min) and the top layer was collected as the sample. To obtain the SER meal samples for testing, the SER meals were defatted with petroleum ether in a Soxhlet extraction unit at 40 °C for 10 h. The oil yield was calculated as follows:Y_i_ (%) = (44–47.35 • X_i_)/(X_i_ + 1)
where Y_i_ represents the oil yield based on fresh rapeseed, 44 signifies the crude oil percentage of fresh rapeseed multiplied by 100, 47.35 corresponds to (1—the crude oil percentage of rapeseed—the water percentage of rapeseed) multiplied by 100, and X_i_ indicates the oil percentage of meal on a dry basis.

### 2.3. Physicochemical Properties (AV, PV, CV, CD, CT, BaP)

The acid value (AV) and peroxide value (PV) of the SER oil were determined using the official methods Ca 5a-40 and Cd 8b-90 of the American Oil Chemists’ Society [14]. The results were expressed in units of mg KOH/g and mmol/kg, respectively.

Conjugated dienes (CD) and conjugated trienes (CT) were determined with reference to IUPAC method II D.23 [15]. A 1% oil–hexane solution was prepared in a 1 cm quartz cuvette, and then the absorbance values were determined at 232 and 270 nm, respectively, using a UV spectrophotometer (UV-1800, Shimadzu, Kyoto, Japan).

The carbonyl value (CV) was determined with reference to [9]. A 0.07–0.12 g oil sample and 2.5 mL benzene were placed into a 15 mL plug test tube. A total of 1.5 mL of trichloroacetic acid solution (4.3 g of solid trichloroacetic acid dissolved in 100 mL of benzene) and 2.5 mL of 2,4-dinitrophenylhydrazine solution were added to the oil samples. The solution was heated to 60 °C for 30 min and then cooled. Then, 10 mL of potassium hydroxide–ethanol solution was slowly added to the above solution and mixed by vortex shaking for 10 min. The absorbance was measured at 440 nm and the zero value was adjusted with a reagent blank. The carbonyl value of the sample was calculated as follows:X_CV_ (meq/kg) = (A × 1000)/(m × 2 × 854)
where A is the absorbance of the sample, and m is mass of the oil sample (g).

Benzo[a]pyrene (BaP) was determined using high performance liquid chromatography (1260 Infinity II, Agilent, Santa Clara, CA, USA) with a fluorescence detector. A total of 0.4 g of SER oil was dissolved in 5 mL of n-hexane. The benzo[a]pyrene molecularly imprinted column was activated sequentially with 5 mL of dichloromethane and 5 mL of n-hexane. The oil solution was transferred to the column and the column was washed with 6 mL of hexane as the liquid level dropped to the column bed. HCl containing 6 mL of dichloromethane and the purified solution was collected into a test tube. The clean-up solution was blown dry with nitrogen at 40 °C, then re-dissolved with 1 mL of acetonitrile, and finally passed through a 0.22 μm microporous membrane for liquid chromatographic determination. An Inertsil ODS-3 column (5 µm × 4.6 nm × 250 mm) was used with a column temperature of 35 °C, an injection volume of 20 µL, and a mobile phase of acetonitrile and water (88:12, *v*/*v*) at a flow rate of 1.0 mL/min. The excitation and emission wavelengths of the fluorescence detector were set to 384 nm and 406 nm, respectively. The standard curve was prepared using 0.5–10 mg/kg of benzo[a]pyrene standard solution.

### 2.4. Carotenoids

Total carotenoid content was measured spectrophotometrically in accordance with the British Standard method [6]. A solution of cyclohexane in 5% oil (*w*/*v*) was prepared and the absorbance was measured at 455 nm using a UV-Vis spectrophotometer. Total carotenoids were calculated using a calibration curve plotted using β-carotene as the standard in the concentration range from 2 to 12 µg/mL.

### 2.5. Total Phenolic Content (TPC)

The total phenolic content (TPC) in the SER oil was determined with reference to the Folin–Ciocalteu method [16]. A total of 0.5 g of oil was dissolved in 1.5 mL of n-hexane and extracted with a methanol–water solution (80:20% *v*/*v*). The lower aqueous phase of the extract was collected by centrifugation at 5000 rpm for 10 min. After the reaction of 1 mL of the extract with 5 mL of Folin’s reagent (10%, *v*/*v*) for 3 min, 4 mL of Na_2_CO_3_ (7.5%, *m*/*m*) was added to the reacted solution for 1 h of reaction in a dark room with the absorbance being measured at 765 nm. A standard curve was plotted using gallic acid as the standard and the values were expressed as mg GAE/kg.

### 2.6. Tocopherols

The tocopherol content of the samples was quantified by high performance liquid chromatography (HPLC) (LC-20AT, Shimadzu, Japan). A total of 0.5 g of SER oil was weighed and diluted in 5 mL of hexane. A SunFire-C18 normal-phase column (150 mm long, 4.6 mm internal diameter, 5 μm particle size) was used with an injection volume of 20 μL, a measurement wavelength of 298 nm, a column temperature of 30 °C, a flow rate of 0.8 mL/min, and a mobile phase mixture of hexane, tert-butyl methyl ether, tetrahydrofuran, and methanol (90:9.45:0.5:0.05, *v*/*v*).

Tocopherols were identified by comparing the retention times of the SER oil sample peaks with those of the standards. Tocopherols were quantified using external standards (α-tocopherol, β-tocopherol, γ-tocopherol, and δ-tocopherol) dissolved in ethanol (1 to 100 μg/mL). The amounts of tocopherols were expressed as mg/kg of oil.

### 2.7. Radical Scavenging Activity (RSA)

The DPPH radical scavenging activity (RSA) was determined based on the reduction of DPPH radicals [17]. A total of 100 μg of SER oil sample was accurately weighed and diluted to 5 mL with ethanol. A total of 100 μL of SER oil dilution was added to a 96-well plate, and 200 μL of 0.1 mmol/L DPPH ethanol solution was added sequentially, incubated for 30 min at room temperature in the dark, and the absorbance was measured at 517 nm. Similarly, 100 μL of Trolox standard solution at different concentrations (0.01–0.1 mmol/L) were, respectively, added into five tubes containing 200 μL of DPPH solution, incubated for 30 min, and the absorbance was measured at 517 nm to obtain a standard curve. The obtained standard curve was used to calculate the antioxidant activity. The results are expressed as mmol/kg Trolox.

### 2.8. Oxidative Stability Index (OSI)

The oxidative stability index (OSI) of SER oil was determined on an 892 Professional Rancimat apparatus (Metrohm, Herisau, Switzerland). A 3 g sample of oil was used for each test. The induction temperature was 120 °C and the airflow rate was 20 L/h. The induction period (IP) was calculated automatically by the apparatus software (StabNet 1.0 Multi CD).

### 2.9. Fatty Acid Composition (FAC)

Fatty acids were methyl esterified and determined by gas chromatography. Heptadecanoic acid methyl ester at a known concentration was used as an internal standard. The analysis was performed using a gas chromatograph (Agilent 6890N, Agilent Technologies, Santa Clara, CA, USA) equipped with a hydrogen flame ion detector (GC-FID) and DB-FastFAME column (30.0 m × 0.25 nm × 0.25 μm, Agilent J&W, Santa Clara, CA, USA). Nitrogen was used as the carrier gas at a flow rate of 0.58 mL/min. The column temperature was initially maintained at 50 °C for 0.5 min and then increased from 50 °C to 194 °C at a rate of 30 °C/min and from 194 °C to 240 °C at a rate of 5 °C/min. The detector and injector temperatures were set to 250 °C. Fatty acids were identified by comparing retention times (RT) to fatty acid methyl ester (FAME) standard peaks and quantified by comparing peak areas to heptadecanoic acid methyl ester. Fatty acid content was expressed in g/100 g.

### 2.10. Maillard Reaction Products (MRPs)

#### 2.10.1. Browning Index (BI)

The browning index (BI) was referred to the method described by [18]. The absorbance of the oil solution (1:20, oil and chloroform, *w*/*v*) at 420 nm was measured to characterize the non-enzymatic browning index of the oil samples.

#### 2.10.2. 5-Hydroxymethylfurfural (HMF)

HMF was determined by a modified colorimetric method [19]. HMF was extracted by adding 1 g of SER oil to 3 mL of 70% methanol, vortexed for 1 min, and centrifuged at 10,000 rpm for 1 min, after which the supernatant was filtered through a 0.45 μm aqueous membrane. A total of 2 mL of the extract was pipetted into a test tube, and then 2 mL of trichloroacetic acid (12 g/100 mL) and 2 mL of thiobarbituric acid (0.025 mol/L) were added. The mixed solution was heated in a water bath at 40 °C for 50 min. After cooling, the absorbance was measured at 443 nm and zeroed with 70% methanol as blank reagent. The standard curve was prepared using 1–8 μg/mL HMF standard solution.

#### 2.10.3. Free Fluorescent Intermediate Compounds (FIC)

Free fluorescent intermediate compounds (FIC) were determined according to the method described by [20]. Oil-isooctane solution (25 mg/mL) was placed in a 1 cm quartz quadruplex cuvette for fluorescence determination (F-4600, Hitachi, Tokyo, Japan) with the excitation wavelength at 350 nm and the emission wavelength at 440 nm. The FIC value is expressed as a percentage of fluorescence of quinine sulfate solution (0.1 μM/0.1 M H_2_SO_4_).

### 2.11. Total Selenium Content

Total selenium (Se) content was determined by hydride generation atomic fluorescence spectrometry (HG-AFS) (Kylin-s12, Ji Tian, Beijing, China). Approximately 0.5 g of the sample (SER oils, SER meals) was weighed into a conical flask, and 5 mL of HNO_3_-HClO_4_ (8:2, *v*/*v*) was added for cold digestion overnight. The next day, the solution was heated in a hotplate at 170 °C, and HNO_3_ was added continuously until the solution was clear and colorless with white smoke. After cooling, 5 mL of HCl (6 mol/L) was added and heating was continued to reduce Se (VI) to Se (IV) until approximately 2 mL of digest remained. The digest was fixed to 10 mL and filtered through a 0.45 μm aqueous membrane. The concentration of Se in the digest solution was determined using atomic fluorescence spectrometry using hydride generation with 5% HCl as the carrier current, a mixture of 0.35% KOH and 2% KBH_4_ solution as the reducing agent, and a Se lamp as the radiation source. The standard curve was plotted with 2–20 μg/L of Se standard solution. The Se content of SER meals was calculated on a dry basis.

### 2.12. Statistical Analysis

All experiments were performed in triplicate and expressed as mean ± standard deviation. An interactive two-way ANOVA was conducted to evaluate the impact of vacuum level and roasting temperature on the quality and Se content of the SER oil. Pearson correlation analysis and principal component analysis were employed to determine the association between the different indicators. Statistical analysis was carried out using SPSS Statistics 26 (IBM, Armonk, NY, USA), while data visualization was accomplished using Origin 2021 (OriginLab, Northampton, MA, USA).

## 3. Results and Discussion

### 3.1. Oil Yield

Table 1 presents the oil yield of unroasted oil and roasted SER oils (90 to 170 °C for 30 min). The oil yield of unroasted SER oil was 32.38%. The oil yield of VC-roasted SER increased from 31.89 to 35.58% with temperature increasing from 90 to 130 °C and then decreased to 34.18% at 170 °C, while the oil yield of DA-roasted SER increased from 31.71 to 35.13% with the temperature increasing from 90 to 150 °C and then decreasing to 30.01% at 170 °C. The increase in oil yield with the increasing temperature was attributed to the increase in cell wall porosity due to protein denaturation during roasting, while the decrease in oil yield when the VC and DA were roasted above 150 and 170 °C, respectively, could be attributed to the excessive low moisture of SER and excessive destruction of SER cells resulting in a small particle size during pressing, which blocked oil flow channels and adsorbed the SER oil [21]. The VC roasting achieved a maximum oil yield of 35.58% at a lower temperature of 130 °C compared to a maximum yield of 35.14% at 150 °C for DA roasting, which may be attributed to the more effective moisture removal ability of the VC roasting to achieve optimal moisture content for oil extraction from SER at the lower temperature. Similarly, a previous study found that vacuum treatment reduced the absorption of oils by dry materials, resulting in a higher oil yield [22]. The F values indicated that both roasting temperature and vacuum had a significant effect on the oil yield from SER (*p* ≤ 0.005), with the temperature having a greater effect on the oil yield (Table 2).

### 3.2. Physicochemical Properties

#### 3.2.1. Acid Value (AV)

The formation of acid value (AV) is due to the oxidation and hydrolysis reaction of triglycerides in oilseeds. Table 1 shows the AV of DA-roasted and VC-roasted SER oil as well as unroasted SER oil. The AV of unroasted SER oil was 0.14 mg KOH/g. The AV of DA-roasted and VC-roasted SER oils increased slightly from 0.17 to 0.22 mg KOH/g and 0.15 to 0.22 mg KOH/g, respectively, with an increasing temperature from 90 to 170 °C, where the AV of all samples are below the maximum limit of 4.0 mg KOH/g [23]. All AVs in the VC-roasted SER oils were slightly lower than in the DA-roasted SER oils at the same temperature from 90 to 170 °C, probably due to the reduced availability of oxygen in the vacuum atmosphere, which can slow down the oxidation reactions of the oil [24]. The AV showed highly significant positive correlations with temperature, CV, CD, and BI (r = 0.920, 0.873, 0.882, and 0.948, respectively, *p* ≤ 0.005) and positive correlations with CT, TPC, RSA, OSI, HMF, and FIC (r = 0.765, 0.699, 0.765, 0.729, 0.758, and 0.724, respectively, *p* ≤ 0.05), as shown in Table 5.

#### 3.2.2. Peroxide Value (PV)

The peroxide value (PV) is an essential physicochemical indicator of oil quality as it represents the primary oxidation products of the oil. Table 1 shows the PV of unroasted and roasted SER oils. The PV of unroasted SER oil was 0.44 mmol/kg. The PV of VC-roasted SER oils increased with the increasing roasting temperature (90–170 °C for 30 min) in the range of 0.30–0.35 mmol/kg, while the PV of DA-roasted SER oils was even higher at 0.58–1.13 mmol/kg, probably due to fewer auto-oxidation reactions initiated by the free radicals of the VC roasting in the absence of oxygen, and the PV of both unroasted and roasted SER oils was below the recommended maximum limit of 5 mmol/kg for edible oils [23]. In addition, the PV of VC-roasted SER oils was also lower than that of unroasted SER oil because VC roasting inactivates lipoxygenases and reduces moisture which promotes the oxidation of SER oil during the oil pressing process [25]. The F values revealed a significant effect of both roasting temperature and vacuum on the PV of roasted SER oils (*p* ≤ 0.005), while the vacuum had a more pronounced effect (Table 2). Furthermore, PV showed highly significant positive correlations with CD, CT, and RSA (r = 0.815, 0.809, and 0.801, respectively, *p* ≤ 0.005) and with CV, TPC, OSI, and HMF (r = 0.725, 0.766, 0.784, and 0.683, respectively, *p* ≤ 0.05), but negative correlations with total Se (r = −0.836, *p* ≤ 0.005), as shown in Table 5.

#### 3.2.3. Carbonyl Value (CV)

The carbonyl value (CV) reflects the content of carbonyl compounds such as aldehydes and ketones produced in the secondary oxidation process of the oil, which is one of the sensitive indicators of oil thermal degradation. As shown in Table 1, the CV of unroasted SER oil was 1.16 meq/kg. With the increase of roasting temperature from 90 to 170 °C, the CV of VC-roasted SER oils increased from 1.31 to 2.05 meq/kg, while the CV of DA-roasted SER oils increased from 1.39 to 3.12 meq/kg, where the CV of VC-roasted SER oils ranged from 65.71 to 95.15% of that of DA-roasted SER oils at the same roasting temperature. A similar study reported that vacuum frying resulted in less degradation of polyunsaturated fatty acids and a lower concentration of carbonyl compounds compared to conventional frying [26]. CV exhibited a highly significant positive correlation with MPRs BI, HMF, FIC (r = 0.921, 0.910, and 0.850, respectively, *p* ≤ 0.005) but negative correlation with total Se (r = −0.816, *p* ≤ 0.005), as shown in Table 5. Carbonyl compounds generated from the thermal oxidation of unsaturated fatty acids have been established to participate in Maillard-type carbonyl–amine condensation reactions with free amino acids or proteins during roasting, resulting in the formation of Maillard reaction products (MPRs) [27], which may also contribute to the loss of Se.

#### 3.2.4. Conjugated Dienes and Trienes (CD and CT)

Conjugated dienes (CD) and trienes (CT) are indicators of the primary oxidation of oils, resulting from the shift of double bonds during the oxidation of fatty acid esters containing methylene-interrupted polyenes or dienes [28]. The CD and CT values of unroasted SER oil were 1.35 and 0.13, respectively. With increasing temperatures ranging from 90 to 170 °C for 30 min, the CD and CT values of VC-roasted SER oils increased from 1.47 to 1.82 and 0.19 to 0.32, while the CD and CT values of DA-roasted SER oils increased from 1.58 to 2.16 and 0.22 to 0.58, respectively. The CD and CT values indicate that higher temperatures lead to the increased oxidation of SER oils. Similarly, a previous study found that the CD and CT values of black cumin seed oil increased with the increasing temperature below 160 °C and then decreased with the increasing temperature above 160 °C due to the accelerated degradation of hydro-peroxide like structures at high temperatures [5], and the CT and CD values in the cocoa butter samples roasted at 150 °C were higher than those roasted at 130 °C [29]. In addition, the lower CD and CT values for the VC-roasted SER oils compared to the DA-roasted SER oils may be due to the vacuum slowing down the oxidation of double bonds in the SER oils. The F values showed that both the roasting temperature and vacuum had a significant effect on CD and CT, with the vacuum having a greater effect (Table 2). Furthermore, CD exhibited a significant positive correlation with CT, TPC, RSA, OSI, BI, HMF, and FIC (r = 0.961, 0.894, 0.902, 0.926, 0.912, 0.883, and 0.807, respectively, *p* ≤ 0.005), total tocopherol (r = 0.671, *p* ≤ 0.05), while negative correlation with total Se (r = −0.875, *p* ≤ 0.005). Moreover, CT was significantly and positively correlated with TPC, RSA, OSI, BI, HMF, and FIC (r = 0.939, 0.920, 0.960, 0.828, 0.913, and 0.848, respectively, *p* ≤ 0.005), total tocopherol (r = 0.725, *p* ≤ 0.05) while significantly and negatively correlated with total Se (r = −0.805, *p* ≤ 0.005), as shown in Table 5, further indicating the synchronization of lipid oxidation with the Maillard reaction, which may lead to Se loss.

#### 3.2.5. Benzo[a]pyrene (BaP)

Benzo[a]pyrene (BaP) is a potent carcinogenic compound that can be produced during the incomplete combustion of organic compounds, including the roasting of oil seeds. In this study, BaP was not detected in unroasted and roasted SER oils at temperatures ranging from 90 to 150 °C (LOD of 0.5 μg/kg). However, BaP levels in VC-roasted and DA-roasted SER oils were 0.59 and 0.68 μg/kg, respectively, when the roasting temperature reached 170 °C (Table 1). These results suggest that BaP is produced at higher roasting temperatures, which is consistent with previous studies [30]. Furthermore, previous studies have shown that BaP is mainly derived from the pyrolysis reaction of proteins and triglycerides in oilseeds [31]. In VC roasting, the thermal degradation of both proteins and triglycerides was inhibited due to the anoxic atmosphere and less convective heat transfer, so the BaP content of VC-roasted SER oils was slightly lower than that of DA-roasted SER oils at 170 °C. This finding suggests that VC roasting is a promising technique to inhibit BaP levels in pressed SER oils.

### 3.3. Carotenoids

The carotenoid content of SER oil is an important quality indicator of its antioxidant properties and health benefits. As shown in Table 1, the carotenoid content of the unroasted SER oil was 42.87 mg/kg. The carotenoid content of the VC-roasted rapeseed oils gradually increased from 56.86 to 76.48 mg/kg as the temperature increased from 90 to 150 °C and then decreased to 61.93 mg/kg at 170 °C, while the carotenoid content of the DA-roasted rapeseed oils gradually increased from 46.89 to 71.87 mg/kg as the temperature increased from 90 to 130 °C and then decreased to 55.26 mg/kg at 170 °C.

Roasting has a complex effect on the content of carotenoids. On the one hand, the breakdown of pigment–protein binding complexes during heating leads to the release of carotenoids into the oils [5]. On the other hand, the higher temperatures promoted the degradation of carotenoids, resulting in lower carotenoid content. Here, VC roasting retarded the thermal degradation of carotenoids compared to DA roasting, thus preserving a higher carotenoid content. The F values indicate that both the temperature and vacuum are significant factors affecting carotenoid content, with temperature being the dominant factor (Table 2).

### 3.4. Total Phenolic Content (TPC)

The unroasted SER oil had a total phenolic content (TPC) of 151.29 mg GAE/kg. By increasing the temperature from 90 to 170 °C, the TPC of the VC-roasted SER oils increased significantly from 184.74 to 391.96 GAE/kg, while the TPC of the DA-roasted SER oils increased significantly from 188.48 to 967.49 GAE/kg (Table 1). At lower temperatures (90 to 110 °C for 30 min), there was no significant difference (*p* ≤ 0.05) in the TPC between the VC-roasted and DA-roasted SER oils. However, when the temperature was increased from 130 to 170 °C, the TPC of the VC-roasted SER oils was 15.31 to 59.49% lower than that of the DA roasting, which could be attributed to more convective heat transfer at higher temperatures, and that the presence of oxygen in the DA roasting promotes protein denaturation and intensifies the destruction of cell structure, resulting in more phenolics being released from the SER meals into SER oils [32]. This finding may explain the higher radical scavenging activity (RSA) and oxidative stability index (OSI) of DA-roasted SER oils compared to VC-roasted SER oils. The F values indicated a significant effect of both temperature and vacuum on TPC (*p* ≤ 0.005), with temperature having a greater effect (Table 2).

### 3.5. Tocopherols

Vitamin E (i.e., tocopherols and tocotrienols) is a kind of lipid-soluble micronutrient and an essential bioactive component of vegetable oils. Table 3 shows the tocopherol content of the unroasted and roasted SER oils. Three types of tocopherols were detected in SER oils: α-tocopherol, γ-tocopherol, and δ-tocopherol. The contents of α-tocopherol, γ-tocopherol, δ-tocopherol, and total tocopherol in unroasted SER oil were 339.67, 703.36, 27.79, and 1081.25 mg/kg, respectively. Both VC roasting and DA roasting resulted in a slight increase in the levels of α-tocopherol, γ-tocopherol, δ-tocopherol, and total tocopherol in SER oils compared to unroasted SER oil, which could be attributed to the thermal damage of the cell membrane during the roasting process where the elaioplast in SER facilitates the release of tocopherols into the oil [33]. The F values indicate that temperature has no significant effect on α-tocopherol and vacuum has a greater effect on α-tocopherol, with significant effects of both vacuum and temperature on δ-tocopherol, and vacuum has a greater effect on δ-tocopherol, while vacuum and temperature have no significant effect on γ-tocopherol (Table 2), which may be related to the higher stability of γ-tocopherol [34].

### 3.6. Radical Scavenging Activity (RSA)

The radical scavenging activities (RSA) of the unroasted and roasted SER oils are presented in Table 1. The unroasted SER oil exhibited an RSA of 2.42 mmol Trolox/kg, which increased from 2.47 to 3.33 mmol Trolox/kg for the VC-roasted SER oils and from 2.51 to 4.64 mmol Trolox/kg for the DA-roasted SER oils as the temperature increased from 90 to 170 °C. The higher RSA of the DA-roasted SER oils over the VC-roasted SER oils can be attributed to its higher TPC in the oils [18]. Moreover, RSA showed significant positive correlation with BI, HMF, and FIC (r = 0.857, 0.973, and 0.957, respectively, *p* ≤ 0.005) as shown in Table 5, suggesting that the increase in RSA may originate from the release of more antioxidant-active Maillard reaction products (MRPs) in the oils [32]. The F-value indicates a significant effect of both temperature and vacuum on RSA, with the effect of vacuum being greater than that of temperature (Table 2).

### 3.7. Oxidative Stability Index (OSI)

The oxidation stability index (OSI) is a measure of the oxidation stability of oil in the process of storage, heating, and frying by measuring the content of the secondary products during the accelerated oxidation of the oil [35]. Table 1 shows the OSI of unroasted and roasted SER oils. The OSI of unroasted SER oil was 5.46 h. As the temperature increased from 90 to 170 °C, the OSI of the VC-roasted SER oils increased from 5.51 to 7.16 h, while the OSI of the DA-roasted SER oils increased from 5.63 to 11.55 h. The OSI of the oil can be attributed to its content of natural antioxidant components, such as phenols, carotenoids, tocopherols, as well as the antioxidant activity of the Maillard reaction products (MRPs) [36], and VC roasting did not improve the OSI of the unroasted SER oils as much as DA roasting, which may be due to the lower TPC and MRPs than the DA-roasted SER oils. The OSI showed a highly significant positive correlation with BI, HMF, and FIC (r = 0.823, 0.979, and 0.933, respectively, *p* ≤ 0.005), thus further corroborating the contribution of MRPs to the OSI of SER oils.

### 3.8. Fatty Acid Composition (FAC)

The fatty acid composition (FAC) of oil is a crucial indicator of its nutritional value, stability and physical properties [33]. Table 4 shows the FAC of unroasted and roasted SER oils. The seven fatty acids detected in SER oil were palmitic (C16:0), stearic (C18:0), oleic (C18:1n9c), linoleic (C18:2n6c), α-linolenic (C18:3n3c), eicosanoic (C20:1n9c), and erucic (C22:1n9c) acids. The unroasted SER oil contained 3.71 g/100 g palmitic acid, 1.14 g/100 g stearic acid, 61.89 g/100 g oleic acid, 16.01 g/100 g linoleic acid, 7.60 g/100 g α-linolenic acid, 1.30 g/100 g eicosanoic, and 0.50 g/100 g erucic acid. The contents of palmitic, stearic, oleic, linoleic, α-linolenic, eicosanoic, and erucic acids in VC-roasted SER oils (90–170 °C for 30 min) ranged from 3.58 to 3.65 g/100 g, 1.09 to 1.16 g/100 g, 60.21 to 60.54 g/100 g, 15.59 to 15.95 g/100 g, 7.37 to 7.64 g/100 g, 1.28 to 1.31 g/100 g, and 0.47 to 0.50 g/100 g, respectively, while the contents of the seven fatty acids in DA-roasted SER oils ranged from 3.29 to 3.58 g/100 g, 1.09 to 1.17 g/100 g, 56.88 to 60.90 g/100 g, 14.43 to 15.76 g/100 g, 6.81 to 7.53 g/100 g, 1.24 to 1.28 g/100 g, and 0.48 to 0.50 g/100 g, respectively. The MUFA and PUFA contents of unroasted SER oil were 63.69 and 23.61 g/100 g, respectively. The MUFA and PUFA of the VC-roasted SER oils decreased by 0.61 to 2.68% and 0.25 to 2.75%, respectively, while the MUFA and PUFA of the DA-roasted SER oils decreased by 1.59 to 7.96% and 1.36 to 10.04%, respectively (90–170 °C for 30 min), compared to the unroasted SER oil. These results indicate that MUFA and PUFA decrease slightly during roasting due to their thermal oxidation and degradation, and that VC roasting has a protective effect on the UFA of SER oil compared to DA roasting, which may be due to less auto-oxidation of UFA in anoxic atmosphere [37].

### 3.9. Maillard Reaction Products (MRPs)

The Maillard reaction is a non-enzymatic browning reaction between carbonyl and amino compounds that occurs extensively during the thermal processing of food products. Free fluorescent intermediate compounds (FIC) are considered as early products prior to the formation of browning products from the Maillard reaction [20]. The FIC value of the unroasted SER oil was 6.46%. As the temperature increased from 90 to 170 °C, the FIC values of the VC-roasted SER oils increased from 6.81 to 36.37%, while the FIC values of the DA-roasted SER oils increased from 7.32 to 61.12% (Table 1). At the same temperatures from 90 to 170 °C, the FIC values of VC-roasted SER oils were consistently lower than those of the DA-roasted SER oils, by 42.53% and 40.50%, especially at 150 °C and 170 °C, respectively.

Another intermediate product of the Maillard reaction, 5-hydroxymethylfurfural (HMF), has positive effects such as antioxidant, anti-allergic, and anti-inflammatory properties. However, the potential carcinogenic, genotoxic, and mutagenic properties of HMF cannot be ignored [38]. The HMF content of unroasted SER oil was 2.17 mg/kg (Table 1). The HMF content did not change significantly at 90 °C and 110 °C (*p* < 0.05), but increased sharply with temperature from 130 to 170 °C, reaching 3.39 to 8.62 mg/kg for VC roasting and 3.82 to 17.20 mg/kg for DA roasting, respectively. Similar to FIC, at the same temperature from 90 to 170 °C, the HMF values of the VC-roasted SER oils were consistently lower than those of the DA-roasted SER oils, by 42.01% and 49.89%, especially at 150 °C and 170 °C, respectively.

The advanced stage of the Maillard reaction can be characterized by the browning index (BI) which represents the polymeric melanin with strong UV absorption at 420 nm [39]. The BI of the unroasted SER oil was 0.17 (Table 1). Similar to FIC and HMF, as the temperature increased from 90 °C to 170 °C, the BI of the VC-roasted and DA-roasted SER oils increased in the range of 0.20 to 0.27 and 0.20 to 0.28, respectively. At a given temperature of 90 to 170 °C, the BI of the VC-roasted SER oil was slightly lower than that of the DA-roasted SER oil, which further confirms the lighter brown color of the VC-roasted SER oil compared to the DA-roasted SER oil.

The results of FIC, HMF, and BI showed that the amount of MRPs increased with increasing temperature (90–170 °C for 30 min) and that VC roasting was lower than DA roasting, which is the reason for the lower RSA and OSI in the VC-roasted SER oils compared to the DA-roasted SER oils, because MRPs (e.g., HMF and nitrogen-containing macromolecules-melanin-like) have antioxidant effects and protect polyunsaturated fatty acids from oxidation [40].

The inhibitory effect of the vacuum on the Maillard reaction in SER oil during roasting is explained as follows. Firstly, the lack of air in vacuum roasting reduces the convective heat transfer and thus the rate of the Maillard reaction [41]. Secondly, the Maillard reaction was inhibited in the anoxic atmosphere due to the reduction of HMF by up to 50% when inert gases (CO_2_, N_2_) were introduced during roasting [10]. Third, lipid oxidation promotes the Maillard reaction because lipid oxidation products become intermediate MRPs or react directly with Strecker degradation products, which may facilitate the Maillard reaction [27,39].

### 3.10. Total Selenium Content

Table 1 shows the selenium (Se) content in SER oils and SER meals. No Se was detected in unroasted and roasted SER oils (LOD of 1 μg/kg), which is consistent with the findings that edible vegetable oils are ineffective for Se supplementation in humans [42]. The Se content of the unroasted SER meals was 218.92 μg/kg. As the temperature increased from 90 to 170 °C, the Se content in the VC-roasted SER meals decreased from 203.22 to 175.66 μg/kg, while the Se content in the DA-roasted SER meals decreased from 187.86 to 165.17 μg/kg, respectively. Here, the Se loss ratio in the VC-roasted and DA-roasted SER meals ranged from 7.17 to 19.76% and 14.18 to 24.55%, respectively, indicating a better retention effect of VC roasting on the Se content of SER meals compared to DA roasting. The total Se of SER meals had a highly significant negative correlation with SER oils AV, PV, CV, CT, CD, RSA, and BI (r = −0.852, −0.836, −0.816, −0.875, −0.805, −0.830, and −0.869, respectively, *p* ≤ 0.005) and a negative correlation with TPC, OSI, HMF, and FIC (r = −0.729, −0.760, −0.733, and −0.708, respectively, *p* ≤ 0.05) as shown in Table 5, indicating that the Se loss of SER is accompanied by lipid oxidation, Maillard reaction, and the increased oxidative stability of the SER oils during the roasting process.

Seleno-amino acids are the main form of Se in SER meals, and Se losses usually occur due to the transformation of seleno-amino acids into volatile Se compounds via oxidative degradation and the Maillard reaction [7,43]. For the oxidative degradation reaction, the seleno-amino acids in the Se-containing proteins of SER have a much lower oxidation potential than the regular thioamino acids, and therefore these seleno-amino acids are highly susceptible to oxidative degradation to form volatile Se compounds [44]. For the Maillard reaction, seleno-amino acids react with reducing sugars to produce volatile Se compounds such as methylselenol, dimethylselenide (DMSe), and dimethyldiselenide (DMDSe) [7]. Therefore, the vacuum reduction of Se loss in SER is attributed to the inhibition of seleno-amino acid oxidation and the inhibition of Maillard reaction of seleno-amino acids in the anoxic atmosphere.

## 4. Principal Component Analysis

Principal component analysis (PCA) was performed to investigate the effect of roasting (vacuum and temperature) on the quality of the SER oil and the Se content in SER meals. The physicochemical properties, micronutrient composition, FAC, oxidative stability, MRPs in SER oils, and Se content in SER meals were analyzed. The results of PCA are shown as score plots and loadings plots (Figure 1). The first two principal components (PC1 and PC2) explained 79.9% of the total variability, with PC1 accounting for 56.7% and PC2 for 23.2% (eigenvalues > 1). The variables highly correlated with PC1 were AV (0.26202), PV (0.227), CV (0.284), CD (0.289), CT (0.284), TPC (0.280), RSA (0.286), OSI (0.282), TTP (0.208), BI (0.274), HMF (0.280), and FIC (0.269), while total Se (−0.256) was negatively correlated with PC1. On the other hand, carotenoids (−0.205), BaP (0.301), SFA (0.483), PUFA (0.489), and MUFA (0.490) mainly contributed to PC2.

The score plot (Figure 1) shows the three groups based on the roasting temperature and vacuum level. The first group (including unroasted, VC90 °C, VC110 °C, VC130 °C, and DA90 °C) had similar characteristics, including higher total Se and fatty acid content, fewer oxidation products, and lower oxidative stability than other DA and VC roasting conditions. The second group (including DA110 °C, DA130 °C, VC150 °C, and DA150 °C) showed improved oil yield compared to the first group, but lower Se and fatty acid content, more oxidation products, and higher oxidative stability than the first group. The third group (VC170 °C and DA170 °C) had the highest degree of roasting from 90 to 170 °C, with the lowest oil yield, the lowest total Se and fatty acid content, the highest amount of lipid oxidation products and Maillard reaction products, and the highest oxidative stability compared to the first and second groups. In Figure 1A, as the roasting temperature increased, the arrangement of the SER samples gradually moved from left to right, and the VC-roasted samples were always located to the left of the DA-roasted samples at the same temperature. This arrangement demonstrated that the increase in roasting temperature led to an increase in the carotenoid content, total phenolic content, tocopherol content and antioxidant capacity, but resulted in a decrease in fatty acids and the Se content, which were inhibited by the vacuum during the roasting process. The load plot (Figure 1B) showed that FIC, HMF, TPC, and BI were strongly correlated with OSI and RSA, indicating a positive contribution of these indicators to the oxidative stability of SER oils. Meanwhile, the obtuse angle between OSI or RSA and total Se indicated that OSI and RSA were negatively correlated with the total Se content, suggesting that roasting enhances the oxidative stability of SER oils while decreasing the Se content in SER meals. Furthermore, the sharp angle between the total Se content and FAC indicated some synchrony in their reduction during roasting, while the obtuse angle between total Se content and MRPs indicated a strong negative correlation between them. These also imply that lipid oxidation and Maillard reaction contribute to Se loss in SER.

## 5. Conclusions

In this study, the effects of VC roasting and DA roasting on the total Se content of SER, oil yield, MRPs, OSI, and various physicochemical properties of SER oil were investigated. The results showed Se was detected only in SER meals and Se loss in VC-roasted and DA-roasted meals increased with increasing temperature, while VC roasting was superior to conventional DA roasting in terms of Se retention. In addition, VC roasting reduced LOPs (AV, PV, CV, CD, and CT), MRPs (BI, HMF, and FIC) and BaP, increased the content of MUFA, PUFA, tocopherols, and carotenoids of SER oil compared to DA roasting. Meanwhile, VC roasting achieved a maximum oil yield of 35.58% at a lower temperature (130 °C), whereas DA roasting (150 °C) achieved a maximum oil yield of 35.14%. Notably, Se loss was positively correlated with LOPs and MPRs. However, the OSI was lower for VC roasting compared to DA roasting above 130 °C because of the reduction of released total phenols and MRPs. Based on the results, it can be concluded that VC roasting has a promising application in Se retention and oil quality improvement of selenium-enriched oilseeds. Further research is needed to explore the promotion of total phenol release into SER oil and the in-depth mechanisms of Se loss in relation to lipid oxidation and Maillard reaction.

## Figures and Tables

**Figure 1 foods-12-03204-f001:**
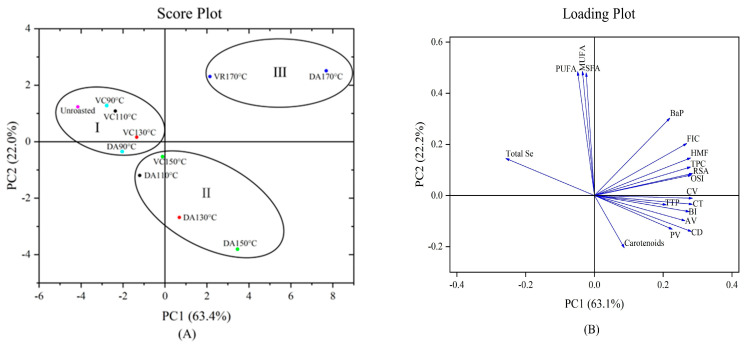
Principal component analysis score plot (**A**) and loading plot (**B**) describing the relationship among different parameters of unroasted and roasted SER oils and total selenium content of SER meals at 90, 110, 130, 150, 170 °C for 30 min. AV = acid value, PV = peroxide value, CV = carbonyl value, CD = conjugated dienes, CT = conjugated trienes, TPC = total phenolic content, RSA = radical scavenging activity, OSI = oxidative stability index, BI = browning index, HMF = 5-hydroxymethylfurfural, FIC = free fluorescent intermediate compounds, BaP = benzo(a)pyrene, SFA = saturated fatty acids, MUFA = monounsaturated fatty acids, PUFA = polyunsaturated fatty acids, TTP = total tocopherol.

**Table 1 foods-12-03204-t001:** Effect of vacuum and dry air roasting on oil yield, chemical properties, carotenoids, total phenols, radical scavenging activity, oxidative stability index, Maillard reaction products, and benzo(a)pyrene content of SER oils and total selenium content of SER.

Parameters	Unroasted (Raw)	DA 90 °C	VC 90 °C	DA 110 °C	VC 110 °C	DA 130 °C	VC 130 °C	DA 150 °C	VC 150 °C	DA 170 °C	VC 170 °C
Oil yield (%)	32.38 ± 0.33 ^d^	31.71 ± 0.11 ^e^	31.89 ± 0.43 ^de^	32.45 ± 0.43 ^d^	31.55 ± 0.14 ^e^	33.51 ± 0.29 ^c^	35.58 ± 0.21 ^a^	35.13 ± 0.31 ^a^	35.13 ± 0.73 ^a^	30.01 ± 0.35 ^f^	34.18 ± 0.12 ^b^
AV (mg KOH/g)	0.14 ± 0.00 ^g^	0.17 ± 0.01 ^e^	0.15 ± 0.00 ^fg^	0.18 ± 0.00 ^d^	0.15 ± 0.01 ^f^	0.20 ± 0.00 ^b^	0.19 ± 0.00 ^c^	0.22 ± 0.01 ^a^	0.20 ± 0.01 ^bc^	0.22 ± 0.00 ^a^	0.22 ± 0.01 ^a^
PV (mmol/kg)	0.44 ± 0.01 ^f^	0.58 ± 0.02 ^e^	0.30 ± 0.02 ^h^	0.66 ± 0.02 ^d^	0.31 ± 0.01 ^gh^	0.77 ± 0.04 ^c^	0.35 ± 0.02 ^g^	0.84 ± 0.01 ^b^	0.35 ± 0.02 ^g^	1.13 ± 0.06 ^a^	0.35 ± 0.01 ^g^
CV (meq/kg)	1.16 ± 0.01 ^f^	1.39 ± 0.04 ^e^	1.31 ± 0.06 ^e^	1.41 ± 0.07 ^e^	1.23 ± 0.10 ^e^	2.11 ± 0.08 ^c^	1.56 ± 0.07 ^d^	2.27 ± 0.10 ^b^	2.16 ± 0.04 ^bc^	3.12 ± 0.16 ^a^	2.05 ± 0.19 ^bc^
CD	1.35 ± 0.06 ^c^	1.58 ± 0.02 ^bc^	1.47 ± 0.06 ^c^	1.63 ± 0.08 ^bc^	1.48 ± 0.06 ^c^	1.82 ± 0.02 ^b^	1.63 ± 0.06 ^bc^	2.16 ± 0.12 ^a^	1.78 ± 0.02 ^b^	2.16 ± 0.48 ^a^	1.82 ± 0.04 ^b^
CT	0.13 ± 0.01 ^h^	0.22 ± 0.01 ^ef^	0.19 ± 0.01 ^g^	0.23 ± 0.00 ^de^	0.19 ± 0.01 ^g^	0.25 ± 0.01 ^d^	0.21 ± 0.01 ^f^	0.50 ± 0.01 ^b^	0.24 ± 0.01 ^d^	0.58 ± 0.01 ^a^	0.32 ± 0.02 ^c^
Carotenoids (mg/kg)	42.87 ± 1.67 ^h^	46.89 ± 1.12 ^g^	56.86 ± 1.55 ^e^	50.85 ± 1.37 ^f^	62.54 ± 0.94 ^d^	71.87 ± 0.40 ^b^	69.08 ± 1.78 ^c^	68.91 ± 1.95 ^c^	76.48 ± 1.76 ^a^	55.26 ± 1.55 ^e^	61.93 ± 0.87 ^d^
TPC (mg GAE/kg)	151.29 ± 2.11 ^h^	188.48 ± 3.16 ^g^	184.74 ± 1.77 ^g^	200.01 ± 4.11 ^f^	199.46 ± 2.93 ^f^	258.81 ± 4.04 ^d^	219.18 ± 2.01 ^e^	491.47 ± 4.61 ^b^	253.50 ± 3.45 ^d^	967.49 ± 8.81 ^a^	391.96 ± 4.83 ^c^
RSA (mmol/kg Trolox)	2.42 ± 0.07 ^f^	2.51 ± 0.10 ^ef^	2.47 ± 0.07 ^ef^	2.77 ± 0.24 ^d^	2.51 ± 0.07 ^ef^	3.05 ± 0.06 ^c^	2.58 ± 0.01 ^ef^	3.34 ± 0.03 ^b^	2.64 ± 0.04 ^de^	4.64 ± 0.02 ^a^	3.33 ± 0.08 ^b^
OSI(h)	5.46 ± 0.07 ^e^	5.63 ± 0.06 ^e^	5.51 ± 0.08 ^e^	5.68 ± 0.12 ^e^	5.58 ± 0.07 ^e^	6.26 ± 0.04 ^d^	5.64 ± 0.04 ^e^	8.35 ± 0.01 ^b^	6.37 ± 0.21 ^d^	11.55 ± 0.52 ^a^	7.16 ± 0.20 ^c^
BI (Abs 420 nm)	0.17 ± 0.00 ^h^	0.20 ± 0.01 ^fg^	0.20 ± 0.02 ^g^	0.22 ± 0.01 ^e^	0.20 ± 0.02 ^ef^	0.25 ± 0.00 ^cd^	0.22 ± 0.01 ^e^	0.26 ± 0.01 ^bc^	0.24 ± 0.00 ^d^	0.28 ± 0.01 ^a^	0.27 ± 0.01 ^b^
HMF (mg/kg)	2.17 ± 0.01 ^g^	2.34 ± 0.05 ^g^	2.32 ± 0.04 ^g^	2.28 ± 0.01 ^g^	2.29 ± 0.01 ^g^	3.82 ± 0.02 ^e^	3.39 ± 0.03 ^f^	7.76 ± 0.23 ^c^	4.50 ± 0.08 ^d^	17.2 ± 0.17 ^a^	8.62 ± 0.02 ^b^
FIC (%)	6.46 ± 0.08 ^i^	7.32 ± 0.40 ^hi^	6.81 ± 0.01 ^hi^	9.15 ± 0.36 ^fg^	8.14 ± 0.00 ^gh^	11.19 ± 0.50 ^de^	10.30 ± 0.14 ^ef^	21.11 ± 1.10 ^c^	11.92 ± 0.20 ^d^	61.12 ± 1.14 ^a^	36.37 ± 2.39 ^b^
Total Se in meal (μg/kg)	218.92 ± 0.55 ^a^	187.86 ± 1.81 ^d^	203.22 ± 1.95 ^b^	182.99 ± 3.88 ^e^	197.53 ± 2.37 ^c^	172.20 ± 1.85 ^g^	196.90 ± 1.14 ^c^	170.97 ± 2.11 ^g^	189.80 ± 1.15 ^d^	165.17 ± 1.81 ^h^	175.66 ± 0.87 ^f^
Total Se in oil (μg/kg)	ND	ND	ND	ND	ND	ND	ND	ND	ND	ND	ND
BaP (μg/kg)	ND	ND	ND	ND	ND	ND	ND	ND	ND	0.68 ± 0.05 ^a^	0.59 ± 0.03 ^b^

Values (mean ± SD, *n* = 3) with similar superscripts in a column do not differ significantly (*p* < 0.05) among roasting treatments. ND = not detected, AV = acid value, PV = peroxide value, CV = carbonyl value, CD = conjugated dienes, CT = conjugated trienes, TPC = total phenolic content, RSA = radical scavenging activity, OSI = oxidative stability index, BI = browning index, HMF = 5-hydroxymethylfurfural, FIC = free fluorescent intermediate compounds, BaP = benzo(a)pyrene.

**Table 2 foods-12-03204-t002:** F values from the ANOVA analysis of the data (vacuum versus roasting temperature) of SER oils and total selenium of SER meals shown in Table 1, Table 3 and Table 4.

	DF	Oil Yield	AV	PV	CV	CD	CT	Carotenoids	TPC	RSA	OSI
Temperature	4	116.91 **	10.91 **	166.78 **	178.09 **	81.67 **	607.54 **	248.50 **	13,186.57 **	226.99 **	269.69 **
Vacuum	1	70.99 **	46.60 **	1606.13 **	115.21 **	121.15 **	1014.27 **	166.00 **	11,420.08 **	254.21 **	268.24 **
Interaction	4	47.43 **	10.86 **	251.00 **	25.74 **	9.00 **	182.23 **	23.90 **	4707.09 **	38.43 **	85.29 **
	DF	α-tocopherol	γ-tocopherol	δ-tocopherol	HMF	FIC	Total Se	SFA	MUFA	PUFA	BI
Temperature	4		3.01 *	5.45 **	2490.02 **	2050.61 **	131.10 **	6.90 **	3.92 *	6.14 **	75.91 **
Vacuum	1	10.82 **		18.86 **	948.08 **	449.51 **	499.12 **	18.98 **	15.98 **	22.88 **	25.83 **
Interaction	4				426.26 **	184.80 **	10.06 **				

* *p* ≤ 0.05, ** *p* ≤ 0.005, DF = degree of freedom, AV = acid value, PV = peroxide value, CV = carbonyl value, CD = conjugated dienes, CT = conjugated trienes, TPC = total phenolic content, RSA = radical scavenging activity, OSI = oxidative stability index, BI = browning index, HMF = 5-hydroxymethylfurfural, FIC = free fluorescent intermediate compounds, BaP = benzo(a)pyrene, SFA = saturated fatty acids, MUFA = monounsaturated fatty acids, PUFA = polyunsaturated fatty acids.

**Table 3 foods-12-03204-t003:** Effect of vacuum and dry air roasting on α-tocopherol, γ-tocopherol, δ -tocopherol, and Total tocopherol of SER oils.

Tocopherols (mg/kg)	Unroasted (Raw)	DA 90 °C	VC 90 °C	DA 110 °C	VC 110 °C	DA 130 °C	VC 130 °C	DA 150 °C	VC 150 °C	DA 170 °C	VC 170 °C
α-tocopherol	339.67 ± 9.21 ^c^	360.22 ± 10.13 ^ab^	363.71 ± 6.84 ^ab^	359.27 ± 6.40 ^ab^	367.39 ± 5.81 ^ab^	359.24 ± 9.63 ^ab^	365.72 ± 6.36 ^ab^	351.35 ± 15.20 ^bc^	374.38 ± 5.09 ^a^	338.10 ± 23.11 ^c^	361.88 ± 4.34 ^ab^
γ-tocopherol	703.36 ± 10.84 ^c^	719.49 ± 29.82 ^abc^	747.62 ± 24.35 ^bc^	744.71 ± 8.90 ^abc^	752.52 ± 12.89 ^ab^	731.06 ± 26.62 ^bc^	746.13 ± 6.54 ^abc^	787.16 ± 38.81 ^a^	749.77 ± 12.16 ^ab^	787.29 ± 45.25 ^a^	758.73 ± 2.34 ^ab^
δ-tocopherol	27.79 ± 1.17 ^d^	33.00 ± 0.86 ^c^	37.25 ± 0.90 ^ab^	36.47 ± 0.61 ^ab^	38.60 ± 0.65 ^a^	36.79 ± 0.88 ^ab^	38.90 ± 2.39 ^a^	34.96 ± 1.44 ^bc^	37.48 ± 1.69 ^a^	38.38 ± 1.48 ^a^	37.89 ± 1.31 ^a^
Total tocopherol	1081.25 ± 19.23 ^b^	1123.75 ± 40.32 ^ab^	1156.38 ± 30.83 ^a^	1147.49 ± 15.57 ^a^	1165.00 ± 19.11 ^a^	1137.65 ± 36.52 ^ab^	1159.57 ± 14.78 ^a^	1190.21 ± 54.90 ^a^	1168.47 ± 18.29 ^a^	1188.46 ± 69.69 ^a^	1164.19 ± 5.11 ^a^

Values (mean ± SD, *n* = 3) with similar superscripts in a column do not differ significantly (*p* < 0.05) among roasting treatments.

**Table 4 foods-12-03204-t004:** Effect of vacuum and dry air roasting on fatty acid composition of SER oils.

Fatty Acids (g/100 g)	Unroasted (Raw)	DA 90 °C	VC 90 °C	DA 110 °C	VC 110 °C	DA 130 °C	VC 130 °C	DA 150 °C	VC 150 °C	DA 170 °C	VC 170 °C
C16:0 (Palmitic)	3.71 ± 0.07 ^a^	3.58 ± 0.03 ^ab^	3.64 ± 0.08 ^ab^	3.49 ± 0.05 ^bc^	3.65 ± 0.03 ^a^	3.49 ± 0.13 ^bc^	3.64 ± 0.12 ^ab^	3.41 ± 0.02 ^cd^	3.60 ± 0.06 ^ab^	3.29 ± 0.09 ^d^	3.58 ± 0.11 ^ab^
C18:0 (stearic)	1.14 ± 0.02 ^a^	1.11 ± 0.07 ^a^	1.16 ± 0.03 ^a^	1.17 ± 0.09 ^a^	1.11 ± 0.09 ^a^	1.12 ± 0.07 ^a^	1.13 ± 0.08 ^a^	1.13 ± 0.03 ^a^	1.13 ± 0.05 ^a^	1.09 ± 0.02 ^a^	1.09 ± 0.04 ^a^
C18:1n9c (Oleic)	61.89 ± 0.73 ^a^	60.90 ± 0.38 ^ab^	61.54 ± 0.29 ^a^	59.68 ± 0.46 ^abc^	61.43 ± 0.32 ^a^	58.64 ± 2.65 ^bcd^	61.36 ± 2.10 ^ab^	57.88 ± 1.61 ^cd^	60.62 ± 1.02 ^ab^	56.88 ± 2.04 ^d^	60.21 ± 1.54 ^abc^
C18:2n6c (Linoleic)	16.01 ± 0.25 ^a^	15.76 ± 0.10 ^ab^	15.92 ± 0.12 ^a^	15.34 ± 0.21 ^abc^	15.93 ± 0.10^6 a^	15.22 ± 0.59 ^bc^	15.95 ± 0.61 ^a^	14.86 ± 0.33 ^cd^	15.64 ± 0.39 ^ab^	14.43 ± 0.37 ^d^	15.59 ± 0.42 ^ab^
C18:3n3c (α-Linolenic)	7.60 ± 0.15 ^ab^	7.53 ± 0.02 ^ab^	7.64 ± 0.19 ^a^	7.33 ± 0.11 ^abc^	7.61 ± 0.06 ^a^	7.24 ± 0.27 ^bc^	7.59 ± 0.30 ^ab^	7.03 ± 0.18 ^cd^	7.46 ± 0.18 ^ab^	6.81 ± 0.24 ^d^	7.37 ± 0.19 ^abc^
C20:1n9c (Eicosanoic)	1.30 ± 0.04 ^a^	1.28 ± 0.04 ^a^	1.29 ± 0.04 ^a^	1.28 ± 0.04 ^a^	1.31 ± 0.04 ^a^	1.27 ± 1.04 ^a^	1.30 ± 1.04 ^a^	1.20 ± 0.04 ^b^	1.29 ± 0.04 ^a^	1.24 ± 0.04 ^ab^	1.28 ± 1.04 ^a^
C22:1n9c (Erucic)	0.50 ± 0.01 ^ab^	0.49 ± 0.01 ^ab^	0.47 ± 0.01 ^b^	0.48 ± 0.01 ^ab^	0.49 ± 0.01 ^ab^	0.50 ± 0.02 ^ab^	0.49 ± 0.03 ^ab^	0.48 ± 0.01 ^ab^	0.49 ± 0.01 ^ab^	0.50 ± 0.01 ^a^	0.49 ± 0.02 ^ab^
SFA	4.85 ± 0.25 ^a^	4.69 ± 0.10 ^abc^	4.80 ± 0.12 ^ab^	4.66 ± 0.21 ^abc^	4.77 ± 0.16 ^ab^	4.61 ± 0.29 ^bc^	4.78 ± 0.31 ^ab^	4.54 ± 0.33 ^cd^	4.73 ± 0.39 ^ab^	4.38 ± 0.37 ^d^	4.67 ± 0.42 ^abc^
MUFA	63.69 ± 0.73 ^a^	62.68 ± 0.41 ^ab^	63.30 ± 0.23 ^a^	61.43 ± 0.47 ^abc^	63.23 ± 0.33 ^a^	60.40 ± 2.72 ^bcd^	63.14 ± 2.18 ^ab^	59.56 ± 1.59 ^cd^	62.40 ± 1.05 ^ab^	58.62 ± 2.04 ^d^	61.98 ± 1.53 ^abc^
PUFA	23.61 ± 0.40 ^a^	23.29 ± 0.12 ^ab^	23.55 ± 0.07 ^a^	22.67 ± 0.32 ^abc^	23.54 ± 0.20 ^a^	22.46 ± 0.86 ^bc^	23.53 ± 0.91 ^a^	21.89 ± 0.51 ^cd^	23.10 ± 0.57 ^ab^	21.24 ± 0.61 ^d^	22.96 ± 0.61 ^ab^

Values (mean ± SD, *n* = 3) with similar superscripts in a column do not differ significantly (*p* < 0.05) among roasting treatments. SFA = saturated fatty acids, MUFA = monounsaturated fatty acids, PUFA = polyunsaturated fatty acids.

**Table 5 foods-12-03204-t005:** Pearson correlation coefficients between the various properties of SER oils and total selenium content of SER meals.

	Temperature	AV	PV	CV	CD	CT	TPC	RSA	OSI	Total Tocopherol	BI	HMF	FIC
AV	0.920 **												
CV	0.849 **	0.873 **	0.725 *										
CD	0.801 **	0.882 **	0.815 **	0.948 **									
CT	0.709 *	0.765 *	0.809 **	0.872 **	0.961 **								
TPC	0.709 *	0.699 *	0.766 *	0.895 **	0.894 **	0.939 **							
RSA	0.744 *	0.765 *	0.801 **	0.907 **	0.902 **	0.920 **							
OSI	0.727 *	0.729 *	0.784 *	0.917 **	0.926 **	0.960 **							
Total tocopherol	0.692 *				0.671 *	0.725 *	0.684 *		0.700 *				
BI	0.945 **	0.948 **		0.921 **	0.912 **	0.828 **	0.798 *	0.857 **	0.823 **	0.643 *			
HMF	0.789 *	0.758 *	0.683 *	0.910 **	0.883 **	0.913 **	0.985 **	0.973 **	0.979 **	0.679 *	0.842 **		
FIC	0.776 *	0.724 *		0.850 **	0.807 **	0.848 **	0.950 **	0.957 **	0.933 **		0.812 **	0.983 **	
Total Se	−0.694 *	−0.852 **	−0.836 **	−0.816 **	−0.875 **	−0.805 **	−0.729 *	−0.830 **	−0.760 *		−0.869 **	−0.733 *	−0.708 *

* *p* ≤ 0.05, ** *p* ≤ 0.005, AV = acid value, PV = peroxide value, CV = carbonyl value, CD = conjugated dienes, CT = conjugated trienes, TPC = total phenolic content, RSA = radical scavenging activity, OSI = oxidative stability index, BI = browning index, HMF = 5-hydroxymethylfurfural, FIC = free fluorescent intermediate compounds.

## Data Availability

The data used to support the findings of this study can be made available by the corresponding author upon request.

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
