# Peer review of "Effect of Vacuum Roasting on Total Selenium Content of Selenium-Enriched Rapeseed, Maillard Reaction Products, Oxidative Stability and Physicochemical Properties of Selenium-Enriched Rapeseed Oil"

_foods, 2023, doi:10.3390/foods12173204_

Round 1
Reviewer 1 Report
This is an interesting study and well presented by authors. Few things that could improve the manuscript.
Introduction: It would be good to include any data for Se levels of canola seed or meal in China and other countries. This is helpful to understand whether the seeds used in the study have high levels than average.
Methods: It is not clear how was the oil yield calculated. Section 2.2 describes pressing and then petroleum ether extraction. How the oil was recovered? Did oil from pressing and solvent extraction combined in the analysis provided including yield and other quality parameters?
Did a control low-Selenium seeds were run in parallel?
Results & Discussion: Was there any color difference for meal and oil in the two methods of roasting?
It is not clear how Se-amino acids get lost during heat treatment. These must be the amino acids in the free form and availble to bind with Se. Most of the amino acids in the seed are from storage protein of the seed which have formed peptide bonds. Some of the residues in the molecule surface may be avialble to form complexes with Se. Without hydrolysing how would these residues become available to bind Se and get broken down during tis temperature? Peptide bond hydrolysis require strong acidic environment and higher energy compared what was provided in this experiment.
Reviewer 2 Report
The manuscript concentrates on investigating the effects of vacuum roasting of rapeseed on oil characteristics, taking into account parameters such as yield or selenium content. The choice of topic is, in my opinion, appropriate and reflects well the needs of the industry, whereas the manuscript lacks an explicit justification for the choice of such a research topic. The choice of research methods is correct, the text is written in the correct language and requires editing, especially in the Conclusions chapter.
General comments:
1. The paper uses a lot of letter abbreviations so please complete the manuscript and their list. This will make the text more readable.
2. What is lacking in the introduction chapter is a comparison of the two drying methods mentioned in terms of technology. Why is dry air roasting used more frequently? Please also cite applications of VC roasting in other food raw materials (sesame, coffee).
3. What is the energy and economic aspect of using VC-roasting in the context of a relatively small increase in oil yield? Will it be cost-effective?
4. The Conclusion chapter definitely needs to be rewritten, as it is difficult to understand in its current form due to the overabundance of values found in the Discussion chapter. Conclusions should summarise all the research results described earlier, but it is not necessary to give specific values with process parameters such as temperature or time. In addition, the Conclusion chapter definitely lacks one-two sentences indicating the potential application of the results obtained.
Reviewer 3 Report
In the manuscript presented for evaluation, the authors roasted selenium-enriched rapeseed. Two roasting methods were used, i.e. conventional dry air (DA) and vacuum roasting (VC). These two processes were carried out at five selected temperatures. The oil was then pressed from the roasted rapeseed and characterised with regard to selected quality parameters. The methods used and the presentation of the results are unobjectionable. Only the following points need to be supplemented:
1. Please enter at least two keywords other than those contained in the title of the manuscript.
2. Please complete the information in section 2.3 regarding the units in which the acid and peroxide numbers were expressed.
3. Please provide in Section 2.7 information on the solvent used to prepare the DPPH solution and the concentration of the solution so prepared.
4. Conclusion needs improvement. It is too long. It should contain basic information on the results obtained and the resulting implications for the future.
